# Comparative Study of Antiviral, Cytotoxic, Antioxidant Activities, Total Phenolic Profile and Chemical Content of Propolis Samples in Different Colors from Turkiye

**DOI:** 10.3390/antiox11102075

**Published:** 2022-10-21

**Authors:** Nazli Boke Sarikahya, Ekin Varol, Gaye Sumer Okkali, Banu Yucel, Rodica Margaoan, Ayse Nalbantsoy

**Affiliations:** 1Department of Chemistry, Faculty of Science, Ege University, Bornova, 35100 Izmir, Türkiye; 2Department of Animal Science, Faculty of Agriculture, Ege University, Bornova, 35100 Izmir, Türkiye; 3Advanced Horticultural Research Institute of Transylvania, University of Agricultural Sciences and Veterinary Medicine, 400372 Cluj-Napoca, Romania; 4Department of Bioengineering, Faculty of Engineering, Ege University, Bornova, 35100 Izmir, Türkiye

**Keywords:** propolis, antioxidant activity, color measuring, Lovibond Tintometer, antiviral, cytotoxicity, total phenolic content, HCA, PCA

## Abstract

Propolis is a valuable natural substance obtained by honey bees after being collected from the bark, resin of trees, plant leaves and mixed with their saliva, and has been widely used for various biological activities. The properties of propolis can vary widely by botanical origin, location of the hives and colony population. It is thought that the color of propolis is one of the main factors determining its acceptability and originates from the flower markers, pollen and nectar of some plants and is directly related to its chemical content. It is important to compare and standardize the colors, chemical content and biological activities of propolis in our country, which has a rich endemic plant diversity. Thus, in this study, the color indexes of 39 propolis samples from different locations in Turkiye were determined by Lovibond Tintometer, for the first time. The color index, total phenolic content, cytotoxic and antioxidant activities relationship of propolis and two commercial propolis samples were also investigated by HCA and PCA. Turkish propolis, which is defined by its color indices, chemical contents and many different activity potentials, such as antioxidant, antiviral and cytotoxic activity, will find use in many fields from medicine to cosmetics with this study.

## 1. Introduction

Propolis is a valuable natural substance that is collected and produced by honey bees *(Apis mellifera*) and it has been widely used for its antioxidant, antibacterial, antifungal and anti-inflammatory properties since the early ages of humanity [1]. The word ‘propolis’ is originated from Greek ‘pro’ which means ‘in front’ and ‘polis’ which means ‘city’, describing this natural product that has a function to guard the entrance to the beehives [2,3]. Propolis is resinous material collected by the honeybees from the bark and resin of trees and various plant sources which is obtained after mixing with their saliva. It is collected, transformed and used by bees to seal holes in their honeycombs, smooth out the internal walls and protect the entrance against intruders. It has a strong odor and can be found in hard, frozen, flexible, sticky and liquid forms depending on the temperature. However, it is found in solid form when first collected from the hive and commonly used in liquid forms by dissolving it in solvents such as ethanol, ether and methanol [4]. In general, it is composed of 50% resin and vegetable balsam, 30% wax, 10% essential oils, 5% pollen and 5% various other substances. More than 800 compounds have been identified in this 5% residue [1,5]. These compounds can be listed as flavonoids [6], terpenes [7], phenolics [8], aldehydes [9], steroids [10], carbohydrates [11], aminoacids [3], aliphatic and aromatic acids and esters [12]. It has wide-spectrum biological effects due to its rich chemical and mineral content. Studies have shown that it has antibacterial, antioxidant [13], antiviral [10], antitumor [14], immunomodulatory [15], anti-inflammatory [16], hepatoprotective [17], cardioprotective [18], neuroprotective [19], antidiabetic [20], regenerative [21], anesthetic [22], antiallergic [23] and biological effects. Parallel to these activities, its effects on inflammatory diseases such as gingivitis, osteoarthritis, mastitis, rhinitis and asthma were also investigated [24]. Propolis is widely used in traditional medicine in many countries due to all these features.

Propolis can be classified depending on its physicochemical properties like color, texture and chemical composition. These properties of propolis can vary widely by botanical origin, location of the hives and colony population. It has a wide color range from brown-yellow, brown-green or brown-red to dark-red. The color of propolis is considered one of the main factors determining its acceptability in accordance with previous reports that have revealed most of its floral markers to be its flavonoids/phenolic compounds which come from the nectar or pollen of specific plants [25]. For instance, Birch and Brazilian Baccharis propolis have a greenish color, while the red propolis from the tropics is reddish [26]. Brazilian propolis is famous all over the world as green propolis characterized by higher levels of phenolic compounds, while the dark and black ones are characterized by mostly triterpenoids. It is produced, predominantly in the southeast of the country, in areas of Cerrado. It is obtained from the apical buds and young leaves of *Baccharis dracunculifolia* (Asteraceae) and has a green color as it contains chlorophyll propolis. The main constituents of Brazilian green propolis are prenylated phenylpropanoids and chlorogenic acid. Flavonoids are also constituents of green propolis, as well as condensed tannins [26,27]. Briefly, the chemical composition of propolis varies depending on the plant, region, season, colony and techniques of collecting propolis; the color, smell, medicinal character and chemical composition of each propolis show differences [28].

According to this scientific backing, in this study we focused on the determination of color, which is one of the determinants of floral origin, chemical content and therapeutic properties of propolis. The color of propolis should be defined in future standardization and a criterion in determining the method of use in apitherapeutic applications. It is important to compare and standardize the colors, chemical content and biological activities of propolis in Turkiye, which has a rich endemic plant diversity. Thus, in this study, the color indexes of 39 propolis samples from different locations in Turkiye were determined by Lovibond Tintometer, for the first time. The color index, chemical, total phenolic contents, antiviral, cytotoxic, and antioxidant activity relationships of these propolis samples and two well-known commercial propolis samples were also investigated by HCA and PCA analyses.

## 2. Materials and Methods

### 2.1. Materials

Propolis samples were supplied by beekeepers from 39 different geographical regions of Turkiye. The exact collection points and locations can be seen in Appendix A and Figure 1.

### 2.2. Color Determination

Color determination analysis was done by Lovibond Tintometer (PFX880). This instrument incorporates calibrated color standards for the particular scale of interest and is operated as a stand-alone instrument. The dried 50–100 mg of the propolis samples were dissolved in ethanol in a 5 mL volumetric flask. The flask was kept in an ultrasonic bath mixed and warmly heated until a clear solution was obtained. Then, the solution was filtered through a 0.45 µm Millipore Millex-HV filter and was placed in the tube of the instrument. The samples were kept at 4 °C until the analysis [30,31].

### 2.3. Chemical Content

The liquid chromatography and high-resolution mass spectrometry (LC-HRMS) methods were developed to analyze the chemical composition of propolis samples from 39 different locations in Turkiye. The identification was performed through the comparison of chromatographic retention times and MS spectra with commercially available standard compounds and the literature findings according to Sarikahya et. al. (2021) [32]. The preparation of samples, chromatographic and optimization conditions for LC-HRMS analysis can be seen in Appendix A.

### 2.4. Antioxidant Capacity and Total Phenolic Content

The antioxidant activities of propolis extracts were determined using 2,2-diphenyl-1-picrylhydrazyl (DPPH) (Sigma-Aldrich, St. Louis, MS, USA) as a free radical, the CUPRAC total antioxidant capacity (TAC) and the ferric-reducing ability (FRAP) of propolis extracts were analyzed by our previous study. Total phenolic contents were also determined according to the Folin–Ciocalteu colorimetric method [29]. All experiments were done in triplicates and all data were shown as mean ± SD.

### 2.5. Cytotoxicity Assay

The cytotoxicity of propolis samples was determined by MTT [3-(4,5-dimethyl-2-thiazolyl)-2,5-diphenyl-2H-tetrazolium bromide)] assay [33]. The test is based on the principle of cleavage of MTT that forms formazan crystals by cellular succinate dehydrogenases in viable cells and doxorubicin was used as a positive control [29]. PC-3 (human prostate adenocarcinoma), MDA-MB-231 (human breast adenocarcinoma), HeLa (human epitheloid cervix adenocarcinoma), A-549 (human alveolar adenocarcinoma) cancer cell lines and normal cell line HEK-293 (human embryonic kidney) were used for assessing cytotoxicity of the propolis samples.

### 2.6. In Ovo Antiviral Activity

Antiviral activities of the 39 different propolis samples were measured as virucidal activity against IBV by *in ovo* [34]. Specific pathogen-free embryo chicken eggs (SPF-ECEs) were purchased from Izmir Bornova Veterinary Control Institute, Turkiye. Favipiravir, used as a broad-spectrum antiviral agent, was purchased from a local pharmacy. While evaluating the antiviral activities of propolis samples, the selection was made according to the content of caffeic acid and some flavonoids such as isosakuranetin, naringenin, rhamnocitrin, diosmetin, and chrysin. 5% DMSO was used as vehicle control and favipiravir was used as positive antiviral control [32]. The protocol for the antiviral test was approved by the Ege University, Local Ethical Committee of Animal Experiment (No: 2020-051).

### 2.7. Statisitics

Hierarchical cluster analysis (HCA) and principal component analysis (PCA) were performed using the Paleontological Statistics (PAST) software (version 4.11, Oslo, Norway) [35]. HCA was performed on a Bray–Curtis similarity with complete linkage. Heatmap and dendrograms were generated using the Euclidean distance based on Ward’s algorithm for clustering [36].

## 3. Results and Discussion

Propolis is one of the most important bee products consumed daily as an immune system supporter and antioxidant agent [35]. It is produced in a wide range of different formulations in the world market. Also, there is a variety of propolis types classified depending on color in each country such as Brazilian green propolis, Portugal red propolis, Egyptian red propolis, etc. [36]. In recent studies, it is determined that the study of the correlations between the parameters examined revealed a significant correlation between the phenolic composition, antioxidant activity and color. The chemical content of the commercially available propolis, such as European poplar propolis and Brazilian green and red propolis, has been studied and standardized [37]. Although Turkiye has a rich flora and significant endemic plant diversity, which can be a good source for propolis, color, antioxidant activity potential and chemical content comparisons have not been studied so far. Therefore, this study is dedicated to further providing information about the color index, chemical composition and total antioxidant capacity, antiviral, cytotoxic activities and phenolic content of 39 propolis samples from different locations in Turkiye (Figure 1, Appendix A) and two well-known commercial propolis samples.

One of the first physicochemical properties used to commercially describe propolis is its color. As we have seen in many studies in the literature, the color of propolis is an important indicator of biological activity and phenolic content. However, there is no official method for propolis color identification and to the best of our knowledge, there is limited literature references concerning the comparative study of color indices, chemical content and biological activity on propolis. In the study conducted by Coelho et al. (2017), colorimetric analyzes of Southeast Brazilian propolis were performed with a Minolta colorimeter CR-400 device, using the CIELAB color system. In the study, it was shown that the Brazilian green propolis has richer phenolic and poorer flavonoid content than the different colored propolis samples collected from the same regions. They also stated that green propolis, which is rich in phenolics, has a higher capacity to capture free radicals and therefore has a higher antioxidant activity [38]. In another color determination study, the colors were determined by the CIELAB system in physicochemical studies on Portuguese propolis by Falcao et al. (2013). It has been shown that even though the hues of the colors are important, the dark green propolis sample has less phenolic content than the light green. Since antioxidant activity is related to phenolic compounds, it is stated that the antioxidant activity capacity of these dark green compounds is low [39]. On the other hand, Machado et al. (2016) performed a comparative study on four different colors of propolis (yellow, red, brown, red) especially focusing on yellow propolis [40].

In this context, this work aims to study the physicochemical parameter color index of 39 propolis samples with various colors (Figure 2) from different locations in Turkiye by Lovibond Tintometer, for the first time. The tintometer is a subtractive colorimeter, which used red, blue and yellow glass standards. Almost everywhere in the world, the Lovibond color scale is nowadays considered an acceptable means of assigning precise color values to edible oils, waxes and fats [31]. In our study red, blue and yellow glass standards in Lovibond Tintometer were used for color detection of 39 propolis samples and two positive controls. The units of red varied from 0.5 to 42.0 units indicating a color change from pale red to dark brown. The red color index of propolis number **31** (Mordogan district of Izmir province) is detected as 42.0 which is the highest red index among forty-one propolis samples. In our previous study, diosmetin is the most abundant chemical substance in propolis **31**. This propolis also showed significant cytotoxicity against the A549 cell line (IC_50_ value of 3.32 ± 0.21 µg/mL). This variation in activity may result from the variability of the non-major compounds in the extract. Another reason for this red color index is that honey bees collect pitch from roads as an adhesive in propolis. If the color and texture are evaluated, it can be said that such a situation may occur in the Izmir propolis sample. The high red color index in propolis in Izmir-Karaburun-Mordogan also corroborates the possibility that it is due to the presence of common Red Pine (*Pinus brutia*) forests in this region. It is determined that increasing the red color index in propolis also increases the hardness of propolis [41]. Furthermore, propolis obtained from pine trees has a lower wax content and a higher resin content. It is seen that the highest yellow index value of 70 is predominant in propolis samples, and this value is found in eight propolis. The lowest yellow index was determined as 2.7 in Rize propolis number **19**. Rize, Çamlıtepe propolis sample contains the highest amount of ellagic acid (12.87 mg/g) as major phenolic acid, and it was determined that the Eastern Black Sea Region, including Rize, Çamlıtepe, is rich in citrus and pine trees. This propolis, which has the lowest yellow index, also has the lowest CUPRAC total antioxidant capacity (TAC) and total phenolic content according to the Folin–Ciocalteu colorimetric method, but the DPPH antioxidant capacity is high. It is clearly seen that the total phenolic content decreases as the yellow color index decreases. These results are also compatible with the literature [38]. The blue color index which is the highest one is detected as 3.4 for both propolis samples from Sivas-Gürün and Tekirdag (**4** and **6**) which is interesting and strengthens the prediction that it originates from *Juniperus excelsa* (*Brown juniper*), which is common in both regions. In addition, the antioxidant capacity of propolis samples **4** and **6** were determined as 83.05% and 83.13%, respectively, as one of the highest values.

Methylquercetin, an antioxidant flavonoid compound, was detected at high levels in propolis collected from Bergama, Izmir (**20**), Hakkari (**22**) and Igdir (**25**), and the yellow index of these propolis samples is varying between 4.2–24.4 in significant value. *Aesculus hippocastanum* L. (horse chestnut) and *Crataegus* L. (hawthorn) trees were widely spread in these regions. These propolis samples also exhibited considerable toxicity on HeLa cells [32]. Quercetin content, which is another antioxidant component, was the highest value with 54.52 mg/g in the propolis sample from Hakkari (**22**). This propolis sample (**22**) contains significantly more phenolic and flavonoid compounds such as chrysin, caffeic acid phenethyl ester, apigenin, acacetin, quercetin, naringenin, rhamnocitrin and diosmetin than any other sample. When it comes to the plant origin of this propolis sample **22**, *Juglans regia* L., *Quercus* spp. L., *Origanum vulgare* L., *Astragalus* L., *Elaeagnus angustifolia* L., *Cotoneaster* spp. Medik and *Morus alba* L plants spread in the region and are responsible for the chemical content. Epigallocatechin, which is the active ingredient of green tea, was determined in Bergama, Izmir (**20**), Muradiye, Manisa (**3**), Trabzon (**12**) and Gumushane (**21**) of which yellow index and antioxidant activity are considerably high. These results establish a direct relationship between yellow color and antioxidant, cytotoxic activity potentials [32] (Appendix A) (Figure 3 and Figure 4).

Recent studies proved that the formation of the most common oxidants in the body, including the superoxide (O_2_^−^•), hydroxyl (OH•), peroxyl (ROO•), alkoxyl (RO•) and hydroperoxyl (HO_2_) radicals, which are collectively known as reactive oxygen species, has been implicated in the oxidative deterioration of food products, as well as in several human pathologies caused by oxidative stress processes. These free radicals are formed via a reduction reaction of molecular oxygen and generate unoccupied electrons, which cause oxidative stress when they are out of equilibrium [42]. Propolis, a rich source of phenolic and flavonoid compounds, can act as an antioxidant with high potentialities in scavenging free radicals associated with various biological activities. The total antioxidant and phenolic capacities in TR equivalents of the same 39 propolis samples were examined by DPPH, CUPRAC, FRAP, and Folin methods and also chemical contents of this propolis were determined in our previous study [29]. The DPPH free radical scavenging model system is a simple method to evaluate the antioxidant activity of compounds in which the purple chromogen radical 2,2-diphenyl-1-picrylhydrazil (DPPH) is reduced to the corresponding pale-yellow hydrazine by the antioxidant component [38]. According to the DPPH results of this study, the antioxidant activity of only seven samples (numbers: **5**, **7**, **30**, **31**, **33**, **34**, **35**) out of 39 samples was found to be below the tested two commercial propolis samples (Brazilian green propolis and Bio-Bee propolis). The highest antioxidant capacity was found in samples **20**, **17**, **37**, **36**, **16** and **2** varied between 84.77 ± 0.02% and 86.17 ± 0.16%. The red and yellow color index of these propolis samples varies between 1.4–3.0 and 5.5–51.0 Lovibond Tintometer, respectively. While the blue color indexes were found as 2.6 and 2.9 only in the propolis samples numbered **16** and **17**, respectively, this value was found to be 0.1 in the others. It is noteworthy that **16** and **17**, which have similar blue color indices, were collected from the same province. The highest benzaldehyde content was determined in the propolis from Artvin (**16**), and it might arise from the presence of *Brosimum alicastrum* Swartz and *Picea orientalis* (L.) trees. Comparing the commercially available propolis samples **40** (Bio-Bee propolis extract) and **41** (Brazilian green propolis), it is seen that the blue color index and antioxidant capacity increased relative to each other. However, it was not concluded that the increase in the blue color index in Turkish propolis directly increased the antioxidant activity. In addition, the variation of propolis color to green with the increase of blue index and yellow color intensity showed that the chemical content of propolis was rich in phenolics. The number of phenolic compounds caffeic acid, ellagic acid, chlorogenic acid, *trans*-4-hydroxycinnamic acid in these green propolis is ranging between 0.84–636.09, 0.01–12.87, 0.01–0.70 and 1.76–62.41 µg/g, respectively (Table 1). The results obtained in the present work are in agreement with the study conducted by Coelho et al. (2017) [38].

Propolis samples numbered **32** demonstrated the highest CUPRAC antioxidant capacity (8.24 ± 0.31 mmol TR g^−1^) and phenolics contents according to the Folin–Ciocalteu method (13.43 ± 0.36 mmol TR g^−1^), whereas **19** showed the lowest total antioxidant capacity. This propolis sample was also found to have the lowest antioxidant capacity with 0.71 (mmol TR g^−1^) CUPRAC and 0.96 (mmol TR g^−1^) compared to Folin–Ciocalteu methods (Table 1). These results can be attributed to the high yellow index, which was 57.0 Lovibond Tintometer in propolis number **32**, compared to 2.7 Lovibond Tintometer in propolis sample number **19**. It is concluded that the lowest antioxidant capacity and yellow index value were determined in Rize-Çamlıtepe (**19**), which is the sample with the weaker chemical content. It is also anticipated that there is a close relationship between the color of the vegetation, which can be a source of propolis for this region and the antioxidant capacity. The Rize-Çamlıtepe region is covered with broad-leaved forests, and there is a propolis source from the Birch family (Betulaceae), the dominant species of Black Alder (*Alnus glutinosa* subsp. *barbata*). It is determined that as the yellow color index decreases, the hardness and antioxidant activity of propolis decreases [41].

The phenolic and flavonoid compounds are correlated with the cytotoxic activity of propolis. Additionally, other compounds identified in the propolis such as triterpenes and sterols are well-known to be responsible for a variety of infectious diseases such as Alzheimer’s, diabetes, hypertension, obesity and cancer [43]. It has been proven in the literature that propolis has cytotoxic activity against Hep-2 (squamous cell carcinoma cell line), Caco-2 (human colon adenocarcinoma), HL60 (human promyelocytic leukemia), MG63 (human osteosarcoma), A549 (human lung adenocarcinoma cell line), MDA-MB-231 (breast cancer cell line), PANC-1 (human pancreatic cancer cell line), HeLa (epitheloid cervix carcinoma) and MCF7 (breast cancer cell line). However, its activity against cell lines HeLa, MCF7 and A549 stands out [43,44,45]. For this purpose, the cytotoxicity results of different color propolis extracts were discussed through a panel of cancerous and nontumor cells in this study. The results of our previous study exhibited that propolis samples numbered **10** (Mugla), **25** (Igdir), **31** (Izmir-Mordogan), **32** (Bursa) and **38** (Istanbul) had the highest cytotoxicity for HeLa, A549 and PC-3 cancer cell lines (Table 2) (Appendix A) [29]. Propolis extracts with high yellow and red color index were more cytotoxic to HeLa cells followed by A549 cells than other cells. In general, propolis samples with high yellow and red color indexes showed significant cytotoxicity, especially on HeLa cells. The first standout of these is propolis number **10**, which was most active in HeLa cells with an index of 8.9 red and 70.0 yellow. In the literature, we found similar results regarding the cytotoxic effect of propolis against HeLa. They proved that yellow propolis is rich in triterpene and has high cytotoxic activity against human ovarian cancer. However, spectrometric color analysis was not carried out in this study (Machado et al., 2016) [40]. Another study found that green propolis extract exhibited an antagonistic effect with doxorubicin in HeLa cells [43].

According to the PCA (Figure 3), from left to right the 1st quadrant highlights samples **21** and **41** with similar antioxidant capacity and cytotoxicity, along with samples **14–17**, **19–21** and **30** with blue index color and moderate cytotoxic activity, particularly in HeLa and A549. Furthermore, an increased antioxidant capacity with regard to DPPH (%) was noticed compared with the other propolis samples, whereas lower capacity was observed in CUPRAC. The 2nd quadrant emphasized samples **23** and **25** with similar cytotoxicity and antioxidant activities, along with samples **4**, **6**, **18**, **22**, **26**, **36** and **39** which exhibited moderate to low cytotoxic activity. In the 3rd quadrant, the propolis samples (**3**, **8**, **10**, **12**, **13**, **31–33**) had lower activity against HeLa and HEK293, whereas higher activities were noticed in MDA-MB 231 and PC3. Furthermore, these samples presented high red and yellow color indexes. The last quadrants presented the propolis samples with the lowest blue color indexes and relatively low CUPRAC and FRAP activities. Out of these, samples **7**, **28**, **35** and **40** demonstrated increased cytotoxicity against all tested lines (Figure 3).

Many studies in the literature demonstrated that most of the propolis samples taken from the temperate zone showed antiviral activity and it is known that flavonoids and esters of phenolic acids are responsible for this activity [46]. Similarly, all 39 propolis samples in our study showed remarkable inhibition of the virus at a concentration of 1 µg/g (Table 3) (Appendix A). The most effective HA titer inhibition was observed as 64 in sample **9** (Usak), which had a blue color index of 2.6, which was higher than the other samples. It also has the best inhibition of HA titer for 0.1 µg/g decreased the virus activity five-fold in comparison with virus control for the 0.1 µg/g concentration. Studies have shown that certain structures in propolis, such as flavonoids and phenolics, cause antiviral effects on the virus. In parallel with the literature, it has been determined that green propolis sample number **9** is rich in terms of flavonoids such as naringenin, rhamnocitrin and phenolic compounds such as caffeic acid. These molecules inhibit the virus by affecting replication mechanisms of viruses and viral envelopes [47].

To better comprehend the similarities and differences between the propolis samples, a dendrogram of the hierarchical clustering and heatmap was constructed and is presented in Figure 4. The first cluster highlights the propolis samples with red (particularly in 31) and yellow color indexes which were correlated with the antioxidant activities as seen by the increased levels in FRAP, CUPRAC and Folin–Ciocalteu. Conversely, a negative correlation was noticed in the cytotoxicity assay. The following cluster highlights the samples with blue color index which exhibited lower antioxidant activities particularly in DPPH (**7**, **30** and **35**) as seen by the negative correlation. On the contrary, samples **7**, **16**, **21**, **28**, **40** and **41** presented increased cytotoxic activity mainly in HeLa and A549 (Figure 4).

As a consequence, the color index of propolis samples differs according to the plant source. A few studies have been done to determine the color index of propolis in literature [38,39,40]. The paler the color resulted in the lower the phenolic content and the antioxidant capacity [8]. The correlations between the phenolic composition and the color revealed that the darker propolis showed a higher total phenolic content (*p* ≤ 0.05) [39]. It also can be observed that the yellow color was negatively correlated with the phenolic content and with the antioxidant activity (*p* < 0.01) for some propolis samples. Therefore, the yellower and paler the color, the lower the phenolic content and the antioxidant capacity, in accordance with that previously observed. Similarly, another study that determined different Spanish propolis samples showed that the lighter color of Spanish propolis could be due to the collection region that is further to the north than propolis becoming darker as one moves towards the south, due to the differences in the local flora. It also reported a significant correlation between the observed color and the antioxidant activity [8,39].

## 4. Conclusions

The propolis products, which are becoming increasingly important and are used as a dietary supplement these days attract a great deal of attention in the pharmacy, cosmetic, food industries and apitherapy. Various countries have focused on determining the chemical composition and different biological activities of propolis to establish their own standards for propolis. Turkiye is the second biggest honey producer in the world with its annual production of 81.115 tons and provides a convenient apicultural environment in terms of flowers [48]. Therefore, considering that propolis is similarly a bee product, Turkish propolis, which is defined by its color indices, chemical contents and potential for many different activities such as antioxidant, antiviral and cytotoxic activity, will find use in many fields from medicine to cosmetics [49]. These results also may be defined in future standardization and a criterion in determining the method of use in apitherapeutic applications for propolis samples. In addition, this study will be guided in the formation of many scientific and industrial studies.

## Figures and Tables

**Figure 1 antioxidants-11-02075-f001:**
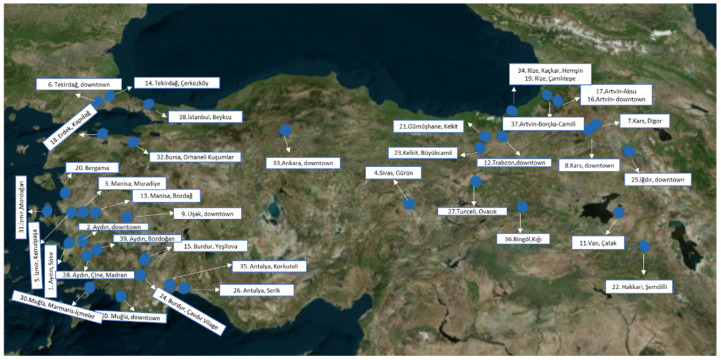
The exact locations of propolis samples collected from Turkiye [29].

**Figure 2 antioxidants-11-02075-f002:**
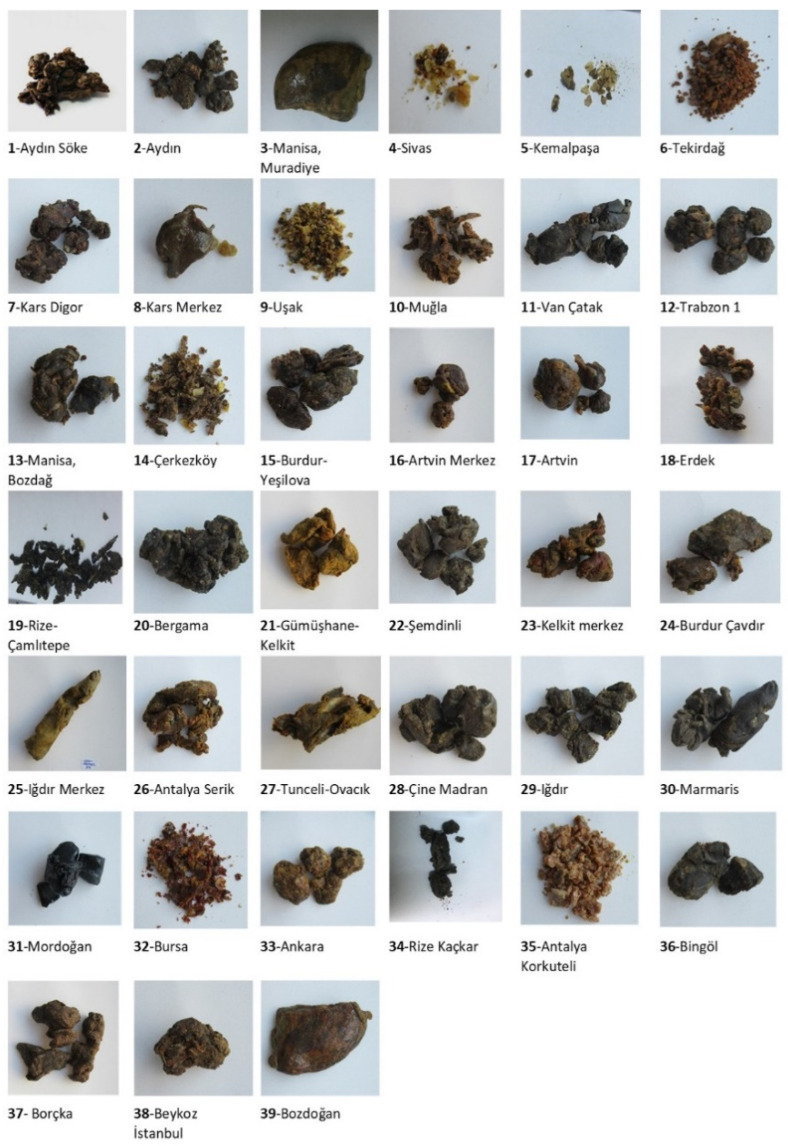
The color variety of the raw propolis samples collected from different geographical locations in Turkiye.

**Figure 3 antioxidants-11-02075-f003:**
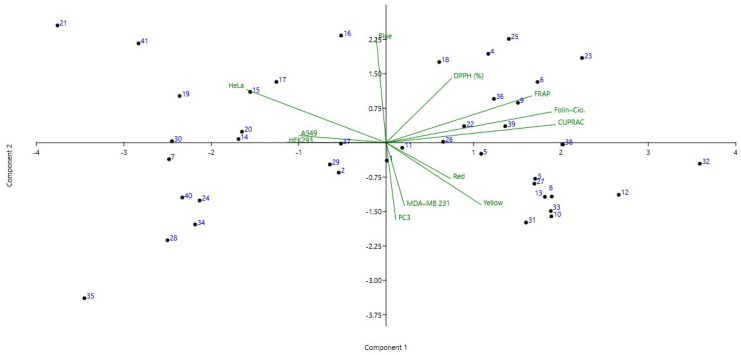
Principal component analysis (PCA) biplot obtained for the propolis samples, color index, antioxidant activities and cytotoxicity. The first two PC explained 65% of the data variance.

**Figure 4 antioxidants-11-02075-f004:**
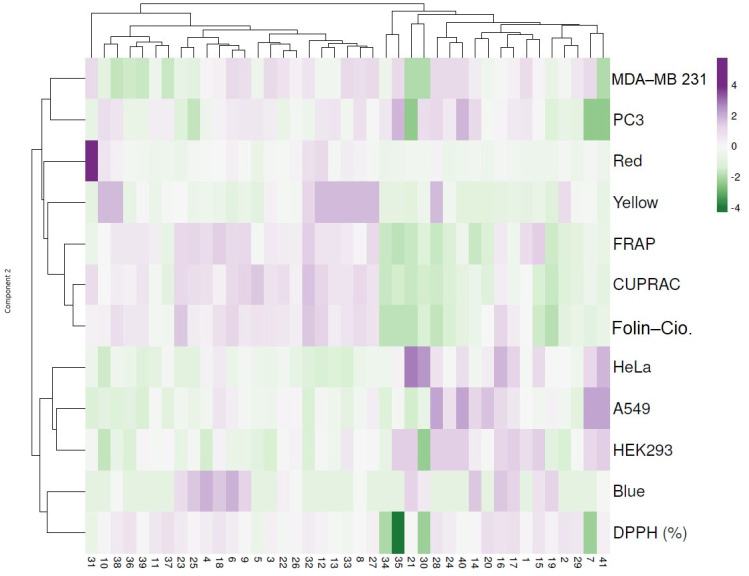
Hierarchical clustering and heat map visualization of the 41 propolis samples based on their color, antioxidant activity and cytotoxicity. Columns indicate the propolis samples and rows the color index, antioxidant activity and cytotoxic assays. Cells are colored based on the quantity in each propolis sample, where purple represents a strong positive correlation and green a strongly negative correlation. The row dendrogram resulted from the correlation between the color index, antioxidant activity and cytotoxic assays; the column dendrogram showed the correlation between propolis samples.

**Table 1 antioxidants-11-02075-t001:** Antioxidant capacity, total phenolic contents and color index of propolis samples collected from Turkiye.

No	COLOR DETECTION			Antiox. Act. ^a^ (%)	Total Phenolic Capacities ^a^ (mmol TR/g)		
	Red	Yellow	Blue	DPPH	CUPRAC	FRAP	Folin–Cio.
**1**	1.6	8.1	0.1	81.48 ± 0.01	4.42 ± 0.17	2.01 ± 0.08	7.39 ± 0.25
**2**	3.0	51.0	0.1	84.77 ± 0.02	1.78 ± 0.35	0.65 ± 0.02	4.49 ± 0.13
**3**	3.0	34.0	0.1	82.22 ± 0.01	6.17 ± 0.21	1.69 ± 0.06	10.03 ± 0.31
**4**	3.5	18.0	3.4	83.05 ± 0.05	6.49 ± 0.24	1.97 ± 0.08	8.56 ± 0.32
**5**	0.5	7.3	0.1	77.49 ± 0.02	8.07 ± 0.09	1.23 ± 0.06	10.49 ± 0.26
**6**	6.2	7.3	3.4	83.13 ± 0.03	7.07 ± 0.08	2.05 ± 0.09	12.33 ± 0.16
**7**	3.0	23.0	0.1	67.96 ± 0.01	3.47 ± 0.10	1.06 ± 0.01	4.39 ± 0.09
**8**	5.8	70.0	0.9	80.35 ± 0.02	7.13 ± 0.19	1.43 ± 0.02	11.42 ± 0.17
**9**	3.6	11.5	2.6	80.83 ± 0.01	7.59 ± 0.21	2.11 ± 0.09	10.00 ± 0.32
**10**	8.9	70.0	0.1	81.27 ± 0.02	4.24 ± 0.03	1.32 ± 0.06	8.56 ± 0.22
**11**	1.5	14.0	0.1	84.22 ± 0.00	3.92 ± 0.09	1.59 ± 0.07	6.86 ± 0.19
**12**	11.5	70.0	0.1	81.72 ± 0.03	7.34 ± 0.08	1.76 ± 0.08	12.42 ± 0.33
**13**	3.0	70.0	0.1	79.77 ± 0.15	6.08 ± 0.11	1.76 ± 0.04	9.39 ± 0.36
**14**	2.4	5.3	2.9	80.67 ± 0.01	3.14 ± 0.13	0.31 ± 0.01	5.56 ± 0.13
**15**	1.8	12.9	1.9	84.67 ± 0.02	1.47 ± 0.07	2.23 ± 0.09	2.09 ± 0.07
**16**	3.1	7.6	2.9	84.87 ± 0.03	5.31 ± 0.13	1.68 ± 0.08	12.28 ± 0.36
**17**	2.6	10.5	2.6	85.76 ± 0.01	3.61 ± 0.09	1.08 ± 0.02	9.14 ± 0.19
**18**	4.4	13.9	2.9	84.51 ± 0.34	5.03 ± 0.11	2.27 ± 0.09	10.65 ± 0.21
**19**	1.4	2.7	2.6	81.72 ± 0.43	0.71 ± 0.01	0.34 ± 0.01	0.96 ± 0.02
**20**	1.4	5.3	0.1	86.17 ± 0.16	2.09 ± 0.04	0.55 ± 0.01	7.57 ± 0.24
**21**	2.4	4.2	2.6	80.68 ± 0.22	0.75 ± 0.01	0.27 ± 0.01	1.18 ± 0.07
**22**	3.8	24.4	1.4	79.68 ± 0.01	6.69 ± 0.22	1.60 ± 0.08	9.85 ± 0.19
**23**	1.2	25.0	2.2	83.59 ± 0.00	7.26 ± 0.24	2.08 ± 0.07	13.53 ± 0.32
**24**	1.3	20.0	0.1	83.09 ± 0.01	1.70 ± 0.09	0.54 ± 0.01	2.91 ± 0.15
**25**	3.3	15.0	2.9	81.19 ± 0.02	6.71 ± 0.23	2.17 ± 0.09	9.30 ± 0.21
**26**	5.7	21.9	0.9	81.34 ± 0.01	5.07 ± 0.19	1.58 ± 0.07	9.10 ± 0.21
**27**	3.2	70.0	0.1	82.32 ± 0.00	6.20 ± 0.34	2.03 ± 0.04	10.38 ± 0.23
**28**	5.5	70.0	0.1	84.40 ± 0.02	0.96 ± 0.02	0.32 ± 0.17	1.65 ± 0.07
**29**	1.5	21.0	0.1	84.28 ± 0.02	2.55 ± 0.05	0.73 ± 0.02	5.28 ± 0.19
**30**	2.5	16.0	1.6	68.03 ± 0.02	1.17 ± 0.03	0.60 ± 0.04	3.12 ± 0.12
**31**	42.0	8.9	0.1	77.00 ± 0.04	6.74 ± 0.18	0.96 ± 0.07	8.37 ± 0.22
**32**	9.9	57.0	0.1	83.26 ± 0.02	8.24 ± 0.31	2.24 ± 0.06	13.43 ± 0.36
**33**	7.5	70.0	0.1	79.09 ± 0.01	6.04 ± 0.34	1.67 ± 0.03	10.16 ± 0.33
**34**	1.2	7.6	0.1	69.08 ± 0.00	0.78 ± 0.01	0.32 ± 0.01	1.16 ± 0.09
**35**	1.4	8.3	0.1	55.98 ± 0.02	1.11 ± 0.03	0.21 ± 0.01	1.24 ± 0.80
**36**	3.4	10.0	0.1	85.27 ± 0.02	5.00 ± 0.21	1.73 ± 0.07	10.01 ± 0.23
**37**	2.0	23.0	0.1	85.56 ± 0.07	2.46 ± 0.05	0.88 ± 0.02	6.55 ± 0.34
**38**	6.1	70.0	0.9	83.87 ± 0.02	5.53 ± 0.17	1.74 ± 0.07	11.11 ± 0.53
**39**	2.7	27.0	0.1	80.99 ± 0.13	6.12 ± 0.20	1.71 ± 0.06	9.85 ± 0.39
**40 ^b^**	2.0	4.9	0.1	79.75 ± 0.07	2.67 ± 0.07	1.02 ± 0.02	4.80 ± 0.07
**41 ^b^**	1.7	10.6	0.9	82.07 ± 0.09	2.68 ± 0.11	0.92 ± 0.02	5.47 ± 0.19

^a^ Sarikahya et al., 2021 ^b^ Propolis products available on the market as a positive control, data are expressed as mean ± SD.

**Table 2 antioxidants-11-02075-t002:** Cytotoxic activity results and color index of propolis samples collected from Turkiye.

No	COLOR DETECTION			CYTOTOXIC ACTIVITY MTT IC_50_ (μg/mL) ^a^				
	Red	Yellow	Blue	MDA-MB 231	HeLa	A549	PC3	HEK293
**1**	1.6	8.1	0.1	47.82 ± 3.58	16.70 ± 3.35	19.05 ± 3.36	35.26 ± 2.88	43.19 ± 5.14
**2**	3.0	51.0	0.1	28.33 ± 0.85	17.04 ± 0.57	16.18 ± 3.40	19.79 ± 1.02	13.25 ± 4.55
**3**	3.0	34.0	0.1	>50	11.63 ± 2.63	14.15 ± 1.97	32.99 ± 5.90	12.21 ± 4.69
**4**	3.5	18.0	3.4	32.86 ± 2.65	20.14 ± 4.27	15.45 ± 4.23	22.44 ± 1.84	11.35 ± 4.90
**5**	0.5	7.3	0.1	24.13 ± 4.90	12.44 ± 5.34	15.06 ± 1.33	35.02 ± 2.18	19.61 ± 5.36
**6**	6.2	7.3	3.4	>50	12.19 ± 3.83	24.38 ± 3.12	33.85 ± 1.19	22.65 ± 2.69
**7**	3.0	23.0	0.1	49.34 ± 0.50	29.67 ± 4.19	>50	ND	46.95 ± 0.46
**8**	5.8	70.0	0.9	46.39 ± 4.57	11.96 ± 0.63	16.32 ± 5.69	38.05 ± 5.81	29.72 ± 4.89
**9**	3.6	11.5	2.6	46.60 ± 5.88	15.75 ± 1.17	17.33 ± 0.24	35.50 ± 5.75	21.99 ± 3.59
**10**	8.9	70.0	0.1	18.08 ± 5.55	1.78 ± 0.01	7.79 ± 0.33	38.55 ± 3.58	8.61 ± 2.62
**11**	1.5	14.0	0.1	23.79 ± 3.43	9.68 ± 0.50	20.05 ± 1.02	32.12 ± 4.71	28.46 ± 2.84
**12**	11.5	70.0	0.1	25.87 ± 2.99	4.46 ± 0.74	17.55 ± 0.54	34.97 ± 1.89	29.22 ± 4.54
**13**	3.0	70.0	0.1	25.82 ± 4.55	5.88 ± 0.76	14.93 ± 1.65	35.78 ± 1.34	25.82 ± 3.73
**14**	2.4	5.3	2.9	38.81 ± 4.74	14.37 ± 1.29	36.68 ± 5.52	40.56 ± 5.10	36.17 ± 5.80
**15**	1.8	12.9	1.9	32.91 ± 2.03	29.14 ± 5.10	16.07 ± 2.45	28.71 ± 4.40	>50
**16**	3.1	7.6	2.9	28.62 ± 6.34	37.47 ± 2.45	34.58 ± 3.75	29.04 ± 5.58	48.04 ± 2.08
**17**	2.6	10.5	2.6	35.31 ± 5.83	32 ± 0.68	28.68 ± 0.87	34.55 ± 3.21	>50
**18**	4.4	13.9	2.9	35.45 ± 5.39	23.20 ± 1.10	28.91 ± 5.22	30.59 ± 2.22	33.09 ± 3.55
**19**	1.4	2.7	2.6	20.95 ± 1.77	17.42 ± 1.20	22.25 ± 4.06	15.39 ± 4.08	19.60 ± 3.58
**20**	1.4	5.3	0.1	25.04 ± 5.56	22.40 ± 1.77	42.18 ± 4.81	26.71 ± 5.82	31.16 ± 3.01
**21**	2.4	4.2	2.6	ND	48.04 ± 0.75	ND	ND	49.60 ± 0.62
**22**	3.8	24.4	1.4	43.90 ± 4.51	14.47 ± 3.37	21.95 ± 0.40	21.20 ± 2.21	30.68 ± 5.30
**23**	1.2	25.0	2.2	19.55 ± 5.67	7.67 ± 1.78	15.38 ± 0.52	15.73 ± 0.60	21.28 ± 1.51
**24**	1.3	20.0	0.1	>50	16.66 ± 3.08	31.84 ± 1.82	33.97 ± 5.80	49.43 ± 1.23
**25**	3.3	15.0	2.9	16.45 ± 3.85	8.59 ± 0.98	11.78 ± 0.50	8.12 ± 0.56	27.74 ± 3.17
**26**	5.7	21.9	0.9	35.32 ± 5.25	10.24 ±0.62	21.31 ± 0.13	25.36 ± 4.97	33.20 ± 0.56
**27**	3.2	70.0	0.1	>50	14.01 ± 0.74	24.82 ± 1.41	28.84 ± 4.79	31.33 ± 1.59
**28**	5.5	70.0	0.1	>50	25.72 ± 2.07	>50	41.83 ± 2.90	>50
**29**	1.5	21.0	0.1	41.20 ± 0.41	14.01 ± 3.17	6.88 ± 2.27	19.28 ± 5.58	29.40 ± 4.08
**30**	2.5	16.0	1.6	ND	44.20 ± 3.68	10.70 ± 3.59	39.03 ± 4.54	ND
**31**	42.0	8.9	0.1	>50	13.21 ± 1.46	3.32 ± 0.21	19.70 ± 0.18	27.39 ± 3.92
**32**	9.9	57.0	0.1	42.14 ± 0.85	6.79 ± 2.12	2.84 ± 0.60	22.84 ± 4.74	15.22 ± 3.44
**33**	7.5	70.0	0.1	>50	4.33 ± 2.09	9.04 ± 3.96	24.27 ± 5.18	39.24 ± 5.54
**34**	1.2	7.6	0.1	23.95 ± 0.34	14.09 ± 0.51	5.21 ± 0.09	35.57 ± 5.39	20.90 ± 2.09
**35**	1.4	8.3	0.1	>50	20.06 ± 5.50	16.72 ± 1.18	>50	>50
**36**	3.4	10.0	0.1	5.86 ± 2.37	11.98 ± 0.38	8.14 ± 0.56	20.29 ± 4.85	14.32 ± 1.13
**37**	2.0	23.0	0.1	4.83 ± 2.99	15.45 ± 2.50	7.29 ± 3.11	32.18 ± 4.96	31.33 ± 5.81
**38**	6.1	70.0	0.9	4.10 ± 1.82	13.98 ± 1.70	4.60 ± 0.44	36.47 ± 5.68	21.01 ± 5.59
**39**	2.7	27.0	0.1	5.38 ± 3.16	6.71 ± 2.16	2.88 ± 0.42	21.23 ± 0.94	32.74 ± 0.36
**40 ^b^**	2.0	4.9	0.1	>50	27.20 ± 2.67	>50	>50	>50
**41 ^b^**	1.7	10.6	0.9	ND	37.64 ± 2.08	>50	ND	>50
**Doxorubicin**				13.14 ± 4.24	1.51 ± 0.38	14.09 ± 2.16	>20	1.10 ± 0.01

^a^ Sarikahya et al., 2021. ^b^ Propolis products available on the market as a positive control, data are expressed as mean ± SD.

**Table 3 antioxidants-11-02075-t003:** Mortality and HA titers of chosen propolis extracts.

Samples	Concentration (μg/g)	Egg Mortality	% Mortality	HA Titer	HA Titer (log2)
Untreated SPF-ECE control		0/4	0%	0	0
Only virus control		0/4	0%	2048	11
Vehicle control (Virus treated with %5 DMSO)		0/4	0%	2048	11
Favipiravir (Positive antiviral agent)	10 25	0/4 0/4	0% 0%	512 256	9 8
Propolis Sample **9** (Usak)	0.1	0/4	0%	64	6
	1	1/4	25%	2	1
Propolis Sample **11** (Catak-Van)	0.1	0/4	0%	128	7
	1	1/4	25%	2	1
Propolis Sample **14** (Cerkezköy-Tekirdag)	0.1	0/4	0%	512	9
	1	1/4	25%	2	1
Propolis Sample **19** (Camlitepe-Rize)	0.1	1/4	25%	256	8
	1	0/4	0%	2	1
Propolis Sample **22** (Semdinli-Hakkari)	0.1	1/4	25%	1024	10
	1	0/4	0%	2	1
Propolis Sample **26** (Serik-Antalya)	0.1	0/4	0%	512	9
	1	0/4	0%	2	1
Propolis Sample **30** (İcmeler-Marmaris)	0.1	1/4	25%	256	8
	1	0/4	0%	2	1
Propolis Sample **37** (Borcka-Artvin)	0.1	0/4	0%	256	8
	1	1/4	25%	2	1

## Data Availability

Not applicable.

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
