# Peer review of "Comparative Study of Antiviral, Cytotoxic, Antioxidant Activities, Total Phenolic Profile and Chemical Content of Propolis Samples in Different Colors from Turkiye"

_antioxidants, 2022, doi:10.3390/antiox11102075_

Round 1
Reviewer 1 Report
This work is comparative study of various Turkish propolis. Although authors describe the biological activities and chemical profiles of the samples, enough data were not reported. Chemical components of propolis depend on the plant origins but there not adequate discussions between the plant origins and chemical components of Turkish propolis. Authors used Brazilian propolis as the reference sample but its plant origin and chemical components are unknown. Authors should provide the other following points also.
#1 Detailed information of the apparatus used for chemical analysis.
#2 Detailed analytical results of the components in each propolis.
#3 All data in Table 2 are not distinguishable. Authors should improve the table.
#4 Some reference names are not abbreviated.
Author Response
The article has been read from the beginning and re-edited in line with the criticisms of all the Reviewers. All changed and added parts are highlighted in the manuscript. Thank you for your valuable comments about our work.
- Chemical components of propolis depend on the plant origins but there not adequate discussions between the plant origins and chemical components of Turkish propolis.
Response: The discussion about the chemical content of propolis samples and their plant origin have been inserted the “Results and Discussion” section as a highlighted. The detailed chemical content of selected propolis samples has also been given in “Sup. Mat.” as Table S3.
- Authors used Brazilian propolis as the reference sample but its plant origin and chemical components are unknown.
Response: Brazilian green propolis is international best seller and most known commercial propolis in the world. The information about plant origin and chemical components of Brazilian propolis have been given in “Introduction section” in detail as a highlighted. The relevant literature was also inserted.
- Detailed information of the apparatus used for chemical analysis.
Response: The detailed information about chemical analysis was inserted to the “Sup. Materials” section.
- Detailed analytical results of the components in each propolis.
Response: Detailed chemical content results of the components in each propolis sample have been inserted as a Table S3 to the “Sup. Materials”.
- All data in Table 2 are not distinguishable. Authors should improve the table.
Response: Table 2 has been improved and split into two tables.
- Some reference names are not abbreviated.
Response: All references have been checked and names of journals have been abbreviated.

Reviewer 2 Report
The ms needs some improvements:
-ClustvisHeat map is not described, also in M&M
-Data concerning the cytotoxicity of propolis samples on some cancer cells compared to normal cell line are of difficult understanding in this way
-Table 2. Antioxidant capacity, MTT and total phenolic content results should be separated and better explained, with also the use of some graphs.
-the authors should also utilize a standard propolis sample, such as a standardized polyphenolic mixtures (for example obtained with Multi dynamic extraction (M.E.D.®) method)
-statistical evaluation should better defined
-discussion of the data is quite long. So, I suggest to focus more on the real applications of the observations.
Author Response
Author's Reply to the Review Report (Reviewer 2)
The article has been read from the beginning and re-edited in line with the criticisms of all the Reviewers. All changed and added parts are highlighted in the manuscript. Thank you for your valuable comments about our work.
-ClustvisHeat map is not described, also in M&M.
Response: The ClustvisHeat map was described in the “Results and Discussion” and “Materials and Methods” part as a highlighted.
-Data concerning the cytotoxicity of propolis samples on some cancer cells compared to normal cell line are of difficult understanding in this way
Response: Table 2 has been improved and the cytotoxicity results have been given in different Table as Table 2. So it will be more understandable.
-Table 2. Antioxidant capacity, MTT and total phenolic content results should be separated and better explained, with also the use of some graphs.
Response: Table 2 has been improved and split into two tables. The cytotoxicity graphs have been also inserted to the “Sup. Materials” as Figure S1.
-the authors should also utilize a standard propolis sample, such as a standardized polyphenolic mixtures (for example obtained with Multi dynamic extraction (M.E.D.®) method)
Response: The standard mixture, which contain known and common phenolic and flavonoid compounds in propolis samples, was already used for these 41 propolis samples from Turkiye and the content was investigated by LC-HRMS. The detailed chemical content results of the components in each propolis sample has been inserted as a Table S3 to the “Sup. Materials”. We also use two kinds of well-known commercial available propolis samples (Brazilian Green propolis and Bio-Bee Turkish propolis extract) and their standard phenolic and flavonoid compounds content was also compared with our propolis samples by same method.
-statistical evaluation should better defined
Response: The statistical evaluation has been inserted to the “Results and Discussion” section as a highlighted.
-Discussion of the data is quite long. So, I suggest to focus more on the real applications of the observations.
Response: The discussion section has been rearranged within the framework of your criticisms.

Round 2
Reviewer 1 Report
The manuscript was appropriately modified.
Reviewer 2 Report
The authors have modifed the ms accordingly